# PHYRE: A New Benchmark for Physical Reasoning

**Anton Bakhtin**     **Laurens van der Maaten**     **Justin Johnson**
**Laura Gustafson**     **Ross Girshick**
Facebook AI Research
{yolo,lvdmaaten,jcjohns,lgustafson,rbg}@fb.com

## Abstract

Understanding and reasoning about physics is an important ability of intelligent agents. We develop the PHYRE benchmark for physical reasoning that contains a set of simple classical mechanics puzzles in a 2D physical environment. The benchmark is designed to encourage the development of learning algorithms that are sample-efficient and generalize well across puzzles. We test several modern learning algorithms on PHYRE and find that these algorithms fall short in solving the puzzles efficiently. We expect that PHYRE will encourage the development of novel sample-efficient agents that learn efficient but useful models of physics. For code and to play PHYRE for yourself, please visit `https://player.phyre.ai`.

## 1   Introduction

Understanding and reasoning about physics is a hallmark of intelligence [9]. Humans can make sense of novel physical situations by reasoning about abstract concepts like gravity, mass, inertia, and friction. For this reason, testing the ability to solve novel physics puzzles has been used to measure the reasoning abilities of human children [7, 53] as well as non-human animals such as capuchin monkeys [52, 54], chimpanzees [40], crows [19, 48], finches [49], and rooks [5]. A key aspect of physical intelligence is *generalization*: after learning to solve a physics puzzle, an intelligent agent should be able to generalize that knowledge and quickly solve related tasks. Robust generalization may set humans apart from other species—prior research showed that four species of non-human primates can learn to solve novel physics puzzles, but struggle to generalize to related tasks [30].

We want to develop artificial systems that can reason and generalize about physics as well as people. However, we hypothesize that in the realm of physical reasoning, present-day machine learning methods will struggle to quickly solve new puzzles. We anticipate that more effective methods may involve fundamental improvements to sample-efficient learning and the ability to learn computationally efficient but useful models of physics.

Towards this goal, we have developed the PHYRE (PHYsical REasoning) benchmark. PHYRE provides a set of physics puzzles in a simulated 2D world. Each puzzle has a goal state (*e.g., "make the green ball touch the blue wall"*) and an initial state in which the goal is not satisfied; see Figure 1. A puzzle can be solved by placing one or more new bodies in the environment such that when the physical simulation is run the goal is satisfied. An agent playing this game must solve previously unseen puzzles in as few attempts as possible. PHYRE was designed to satisfy three main goals:

- **Focus on physical reasoning:** Tasks are as simple as possible but still require nontrivial physical reasoning. Scenes are built only from balls and rectangular bars. Dynamics are deterministic, with only collision, gravity, and friction. Goals are symbolic, so natural language is not required.
- **Focus on generalization:** After training on one set of tasks, we should expect an effective agent to solve new, previously unseen puzzles. The benchmark is structured such that puzzles are split into training tasks and evaluation tasks, and involves two different degrees of generalization.
- **Focus on sample-efficiency:** Our evaluation rewards solving tasks with as few attempts as possible. Methods that master a task only after thousands of attempts will not perform well.

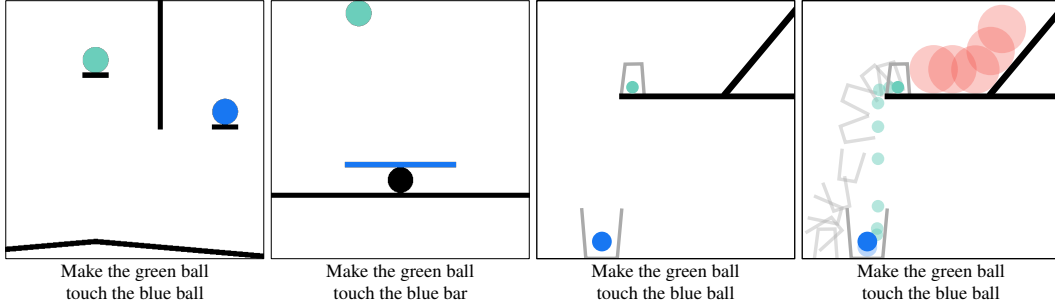

| Make the green ball | Make the green ball | Make the green ball | Make the green ball |
| touch the blue ball | touch the blue bar | touch the blue ball | touch the blue ball |

Figure 1: Three examples of PHYRE tasks (left) and one example solution (right). Black objects are static; objects with any other color are dynamic and subject to gravity. The tasks describe a terminal goal state that can be achieved by placing additional objects in the world and running the simulator. The task in the left-most pane requires placement of two balls to be solved, whereas the others can be solved with one ball. The right-most pane illustrates a solution (red ball) and the solution dynamics.

Figure 1 shows three examples of PHYRE tasks. Each task comprises static and dynamic objects in a 2D environment and a goal description. Upon looking at these examples, you, the reader, will likely form an intuitive hypothesis for how to solve each problem. If your first attempt were to fail, you would likely be able to use your observations of what happened to refine your attempt into a successful solution. PHYRE encourages the development of learning agents with similar abilities.

## 2    Related Work

PHYRE is related to prior work on intuitive physics, visual reasoning, and learning in computer games and simulated (robotics) environments. It was developed concurrently with the Tools game [1].

**Intuitive physics.** Foundational work in cognitive psychology suggests that people reason about the physical world using simplified intuitive theories [31–33]. Early computational instantiations of this framework used probabilistic models over physical simulations [3, 61], while more recent methods use feedforward neural networks trained to make pixelwise [11, 24, 34, 60] or qualitative [12, 26, 27, 34, 61] predictions about the future, sometimes in conjuction with simulation [18, 57, 58]. Many methods are evaluated on the constrained task of predicting whether a 3D stack of blocks will topple; some recent studies instead ask models to determine whether videos of more complex scenes are physically plausible [39, 41]. In contrast, PHYRE provides a suite of goal-driven tasks to test intuitive physical understanding: rather than evaluating intermediate tasks like future prediction or stability modeling, PHYRE requires agents to intervene in the scene to achieve a desired end goal.

**Visual reasoning.** Work on visual reasoning dates to SHRDLU [56] which probed scene understanding using natural language; more recent benchmarks require systems to answer natural-language questions about images [2, 20]. Recent methods use neural networks to extract sub-symbolic image representations, which are used in subsequent reasoning modules [16, 17, 21, 29, 37, 43]. Like PHYRE, these tasks require reasoning about the interactions of multiple objects in a scene; however unlike PHYRE they assume a static world, and do not require reasoning about world dynamics.

**Learning in computer games.** Computer games often involve complex 2D and 3D environments and require agents to possess some level of physical understanding [35, 46]. For instance, Atari games such as Pong involve precise positioning of a paddle based on observed ball dynamics [35]. The main difference between prior work in computer games and our work is that PHYRE requires the agent to learn a single model to solve a wide range of different tasks rather than a specialized model for each task. Moreover, in contrast to most work in computer games, PHYRE requires the agent to learn in a sample-efficient manner, penalizing agents that require many samples to learn.

**Learning in simulated (robotics) environments.** A range of prior work studies learning in simulated (robotics) environments for self-driving cars [10], humanoid robotics [50], or navigation tasks [44, 59]. In contrast to PHYRE, these prior studies focus on agents operating in realistic non-deterministic 3D environments in which the world is not fully observed, which hampers systematic study of the reasoning capabilities of the agent. By contrast, PHYRE takes inspiration from CLEVR [20] and limits the complexity of the environment, facilitating more systematic analysis of reasoning abilities.

# 3 PHYRE Physical Reasoning Benchmark

The PHYRE environment is a two-dimensional world that simulates simple deterministic Newtonian physics. There is a constant downward gravitational force and a small amount of friction. All *bodies* are non-deformable and are either *static* or *dynamic*, distinguished by color. Static bodies remain at a fixed position and are not influenced by gravity or collision, while dynamic bodies move in response to these forces. All bodies come from a small *vocabulary*[1] and vary in scale, location, and orientation.

A PHYRE *task* consists of an *initial world state* and a *goal*. The initial world state is a pre-defined configuration of bodies. The goal is a (subject, relation, object) triplet identifying a *relationship* between two bodies that the agent needs to achieve when the simulation terminates. At present all tasks use a single relation, `touching` for at least 3 seconds, which we found sufficient for developing a diverse set of tasks. The environment can be extended in the future to include additional relationships.

The agent aims to achieve the goal by taking a single *action*, placing one or more new dynamic bodies into the world. Bodies placed by the action may not extend beyond the world boundaries or intersect other bodies; such actions are rejected by the simulator as *invalid*. After the action is taken, the simulator runs until the goal is reached or until a time limit elapses, whichever happens first. The agent cannot perform additional actions while the simulator runs. Once the simulation is complete, the agent receives a binary *reward* indicating whether the goal was achieved, and gains access to observations of the intermediate world states produced by the simulator. If the goal was not achieved, the world resets to its initial state and the agent tries again, possibly informed by its prior attempts.

The full world state, comprising exact positions and orientations of bodies as well as their masses and velocities, is not revealed to agents since human observers cannot directly perceive such values from their environments. Instead, the agent receives coarser initial and intermediate world states as *observation images* which rasterize the world to a 256×256 grid. Each grid cell takes one of seven values specifying whether that location is a (1) dynamic goal object, (2) static goal subject, (3) dynamic goal subject, (4) static confounding body, (5) dynamic confounding body, (6) body placed by the agent, or (7) background. With only one relation, the colors in the initial observation encode the goal, eliminating the need for natural-language goal specification or grounding. Figure 1 shows three PHYRE tasks with goals written in natural language solely for the convenience of the reader.

Without any restrictions on the action space, for example on the body types, their properties, and the number of bodies that may be placed, the action space is large and complex. We therefore define two restricted action *tiers* for the current benchmark, which we describe next. After research progresses on these tiers, more complex ones may be added to the benchmark.

## 3.1 Benchmark Tiers

This work studies two benchmark tiers of increasing difficulty. A *tier* comprises a combination of: (1) a predefined set of all actions the agent is allowed perform and (2) a set of tasks that can be solved by at least one action from this action set. The two tiers we developed for this study are:

- **PHYRE-B.** Action set containing all valid locations and radii for a single ball (3D; continuous).
- **PHYRE-2B.** Action set containing all valid pairs of two balls (6D; continuous).

The two tiers each contain 25 task *template*s. A task template defines a set of related tasks that are generated by varying task template parameters (such as positions of initial world bodies). All tasks in the same template share a common goal, but have different initial world states. Each template defines 100 such tasks. Task templates are used to measure an agent's generalization ability in two settings. In the **within-template** setting, an agent trains on a subset of tasks in the template and is evaluated on the remaining tasks within that template. To measure **cross-template** generalization, test tasks are selected exclusively from templates that were not used for training. Our criteria for task design, additional analysis, and visualizations of tasks are provided in the supplement.

## 3.2 Learning Setting

Because the agent can only perform a single action to solve a PHYRE task, PHYRE is similar to a *contextual bandit* setting [23, 25]. PHYRE differs from traditional contextual bandit settings in two

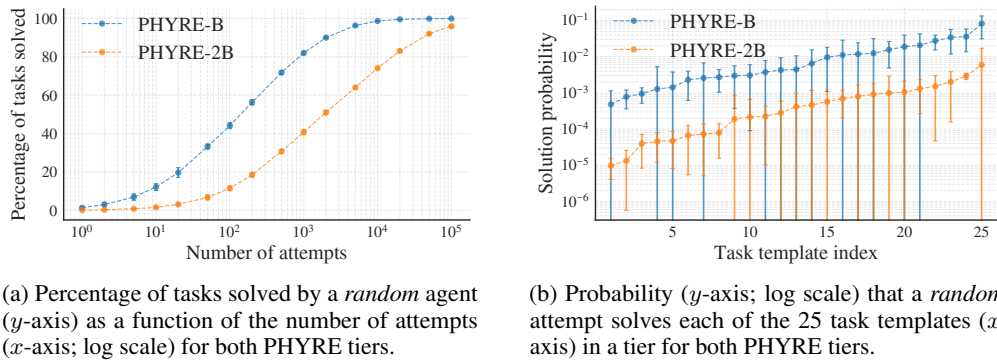

(a) Percentage of tasks solved by a *random* agent ($y$-axis) as a function of the number of attempts ($x$-axis; log scale) for both PHYRE tiers.

(b) Probability ($y$-axis; log scale) that a *random* attempt solves each of the 25 task templates ($x$-axis) in a tier for both PHYRE tiers.

Figure 2: PHYRE complexity analysis. Values are averaged over 10 runs over all tasks in the tier; error bars indicate one standard deviation. Two-ball tasks are much harder to solve by chance than single ball tasks. Each tier contains a spectrum of task difficulty with respect to random guessing.

main ways: (1) it has an offline training phase that precedes the online learning testing phase and (2) the agent receives privileged information [51] in addition to the binary reward signal, *viz.*, it has access to observations of intermediate world states produced by the simulator on previous attempts.

In the **training phase**, the agent has access to the training tasks and unlimited access to the simulator. The agent does not have access to task solutions, but can use the simulator to train models that can solve tasks. Such models may include forward-prediction or action-prediction models.

In the **testing phase**, the agent receives test tasks that it needs to solve in as few *attempts* (queries to the simulator) as possible. After each attempt, the agent receives a binary reward and observations of intermediate world states. The agent can use this information to refine its action for the next attempt. Some actions may be invalid, *i.e.*, correspond to an object that overlaps with other objects. In such cases, we neither give the agent any reward nor count this attempt toward the query budget. The agent receives access to all test tasks at once, allowing it to choose the order in which it solves tasks.

**Performance measure.** We judge an agent's performance by how efficiently it solves tasks in the testing phase. We characterize efficiency in terms of the number of actions that were attempted to solve a given task; fewer attempts corresponds to greater efficiency. We formalize this intuition by recording the cumulative percentage of test tasks that were solved (the *success percentage*) as a function of the number of attempts taken per task. To compare the performance of agents on PHYRE, we plot this success-percentage curve. We also compute a performance measure, called **AUCCESS**, that aggregates the success percentages in the curve via a weighted average. To place more emphasis on solving tasks with fewer attempts, we consider the range of attempts $k \in \{1, \ldots, 100\}$ and use weights $w_k = \log(k+1) - \log(k)$, yielding AUCCESS $= \sum_k w_k \cdot s_k / \sum_k w_k$, where $s_k$ is the success percentage at $k$ attempts. The relative weight of the first 10 attempts in the AUCCESS measure is $\sim$0.5: agents that need more than 10 attempts cannot get an AUCCESS score of more than 50%. This encourages the development of sample-efficient agents. AUCCESS is equivalent to the area under the success-percentage curve formed by replacing the discrete samples with a piecewise constant function and placing the number of attempts on a log scale.

### 3.3 Analysis

To assess the difficulty of the tasks in both PHYRE tiers, we measured what percentage of PHYRE tasks can be solved by an agent that randomly samples actions from the action space. Figure 2a shows the percentage of tasks ($y$-axis) that this random agent solves in at most $k$ attempts ($x$-axis), averaged over 10 runs on all PHYRE tasks. The figure reveals that tasks vary greatly in difficulty level: a few tasks can be solved by a random agent in just a few attempts, whereas other tasks require thousands of attempts to be solved. The figure also shows that tasks in the PHYRE-2B tier are, on average, harder than those in PHYRE-B because the action space in that tier has more degrees of freedom.

We designed the PHYRE tasks such that, on average, it takes a random agent no more than 10,000 attempts to solve task in the PHYRE-B tier and no more than 100,000 attempts to solve a task in the PHYRE-2B tier. Figure 2b illustrates this by displaying the average probability that a random attempt

solves a task for each of the 25 task templates in both PHYRE tiers. In line with the previous analysis, the figure also shows that tasks in PHYRE-2B are substantially harder than those in PHYRE-B.

## 4 Experiments

We conduct experiments to obtain baseline results for within-template and cross-template generalization on the PHYRE benchmark. Experiments are performed separately on each tier. Code reproducing the results of our experiments is available from `https://phyre.ai`.

### 4.1 Baseline Agents

We experiment with five baseline agents that rank actions given an observation of the initial state (recall that the observation encodes the goal): (1) a random agent, (2) a non-parametric agent, (3) a deep Q-network [35], and (4-5) counterparts to (2) and (3) that update the agent online during testing.

**Random agent (RAND).** This agent does not perform any training and instead samples actions uniformly at random from the 3D or 6D (depending on the tier) action space at test time.

**Non-parametric agent (MEM).** At training time, this agent generates a set of $R$ random actions and uses the simulator to check if each of these actions can solve each of the training tasks. For each action $a$, the agent computes $p_a$: the fraction of training tasks that the action solves. The agent then sorts the $R$ actions by $p_a$ (highest to lowest), and tries them in this order at test time. This agent is non-parametric because it uses a list of "memorized" actions at test time.

In the *cross-template setting*, the test tasks come from previously unseen task templates and this simple agent cannot relate them to tasks seen during training. It therefore uses the same action ranking for all tasks and ignores the observation of the initial state. In the *within-template setting*, each test task comes from a task template that was seen during training. In this case, we give the agent access to the task template id for each test task. The agent maintains a per-task-template ranking of the $R$ actions. The same set of actions is shared across all templates; only the ranking changes. The set of actions attempted on each task may vary because invalid actions are ignored; see Section 3.2.

**Non-parametric agent with online learning (MEM-O).** This agent has the same training phase as the non-parametric agent, but continues to learn online at test time. Specifically, after finishing each test task (either successfully or unsuccessfully), the agent updates $p_a$ based on the reward received for each action $a$ in the subset of the actions it attempted. The updated ranking is used when the next task is attempted. Such online updates are beneficial, in particular, in the cross-template setting because they allow the agent to learn something about the tasks in the previously unseen templates. We use cross-validation to tune the relative weight of the update on each train-val fold (see Section 4.2).

**Deep Q-network (DQN).** As before, the DQN agent collects a set of observation-action-reward triplets by randomly sampling actions and running them through the simulator. The agent trains a deep network on the resulting data to predict the reward for an observation-action pair. Following [4], we train the network by minimizing the cross-entropy between the soft prediction and the observed reward. During training, we sample batches with an equal number of positive and negative triplets.

Our network comprises: (1) an *action encoder* that transforms the 3D or 6D (depending on the tier) action representation using a multi-layer perceptron with a single hidden layer; (2) an *observation encoder* that transforms the observation image into a hidden representation using a convolutional network (CNN); and (3) a *fusion module* that combines the action and observation representations and makes a prediction. Our action encoder is a MLP with a single hidden layer with 512 units and ReLU activations. Our observation encoder is a ResNet-18 [13]. For the fusion module, we follow [37] and use the action encoder to predict a bias and gain for each channel in the CNN. The output of the action encoder thus contains twice as many values as there are channels in the CNN at the fusion point. To expedite action ranking, we fuse both models before the last residual block of the CNN. We tried other fusion points but did not observe performance differences (see supplemental material).

The observation of the initial state is a 256×256 image with one of 7 colors at each pixel, which encodes properties of each body and the goal. We map this observation into a 7-channel image for input to the CNN; each colored pixel in the image yields a 7D one-hot vector. Following common practice, the network is trained end-to-end using stochastic gradient descent with the Adam optimizer [22]. We anneal the learning rate to 0 using a half cosine schedule without restarts [28].

**Deep Q-network with online learning (DQN-O).** Akin to MEM-O, this agent uses rewards from test tasks to perform online updates. After finishing a test task, the agent performs a number of gradient descent updates using examples obtained from that task. The updated model is then used for the next test task. The number of updates and corresponding learning rate are set via cross-validation.

**Contextual bandits.** While PHYRE is a contextual-bandit setting, we found that contextual bandits (CBs) do not work well on our complex observation and action space. Most CBs model the expected reward given the context and action using linear models [6, 8, 25], Gaussian processes [47], or deep neural networks [42]. Linear models do not yield useful context representations (which are observation images). Gaussian processes require a reasonable kernel function on the observation image space, which is difficult to define. Methods based on deep neural network seem more suitable. We tried to use the implementation from [42][2], but were unable to train the model once we replaced the shallow MLP used in [42] by a CNN that is better suited for image encoding. In addition, CBs generally assume a fixed (usually small) number of arms without a similarity metric between the arms, which is problematic for PHYRE tasks: when reasonably discretized, the number of arms in PHYRE-B is $\sim 10^6$. Moreover, without considering similarity between actions, agents try various non-working arms in the same region of the action space without diversifying them (see Section 4.4).

**Policy learners.** We faced similar issues with policy learners such as PPO [45] and A2C [36]. While we were able to factorize the action space over each dimension and use continuous action spaces, we were unable to train models that outperform our random baseline due to poor training stability.

## 4.2   Experimental Setup

We measure success percentage and AUCCESS on PHYRE using the learning setting of 3.2. To make results reproducible and allow fair comparisons between agents across studies, PHYRE provides:

- A fully deterministic environment: agents always produce the same result on a task.
- A process that deterministically splits the tasks into 10 *folds* containing a training, validation, and test set. As a result, agents are always compared on exactly the same task splits. Task splits are available for both tiers and both generalization settings (within-template and cross-template).

To avoid overfitting on test tasks, hyperparameter tuning is only to be performed based on the validation set: *we discourage tuning of hyperparameters based on test task performance.* For results on the test set, we use these tuned hyperparameter and train agents on the union of the training and validation sets. To compare agents, we use the non-parametric Wilcoxon signed-rank test for median difference [55] with one-sided null hypotheses and $p = 0.01$. We use this test as it does not have a normality assumption, is efficient with small sample sizes, and works with relative values on each fold instead of absolute values. To facilitate comparisons with our baselines, we provide our AUCCESS scores on all 10 folds in the supplementary material.

At test time, all agents (except the random agent) rank the same set of 10,000 actions on each task and propose the highest-scoring actions for that task as solution attempts. The MEM(-O) agents were trained on the same 10,000 actions. The DQN(-O) agents were trained on 100,000 actions per task.[3] All agents are permitted to make up to 100 attempts per task. This fact subtly implies that when computing the success percentage at $k < 100$ attempts, online agents will have learned from up to 100 (not $k$) attempts per task; this pragmatic choice makes the benchmark computationally tractable as otherwise online agents would need to be re-run for every value of $k \in \{1, \ldots, 100\}$.

## 4.3   Main Results

Figure 3 presents success-percentage curves for all five agents on both PHYRE tiers (-B and -2B) in both generalization settings (within-template and cross-template): the curves show the percentage of tasks solved as a function of the number of solution attempts per task, and are computed by averaging over all 10 folds in PHYRE. Table 1a presents the corresponding mean AUCCESS (and its standard deviation). The results are in line with the trends observed in Section 3.3: the within-template setting is much easier than the cross-template setting for all (non-random) agents. As forecasted, the two tiers also have different difficulty characteristics. In the cross-template setting, the best agent,

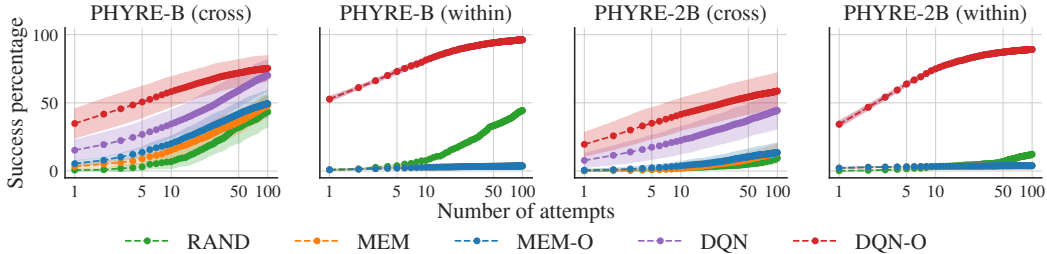

Figure 3: Percentage of solved tasks (success percentage) as a function of the number of attempts per task of five agents on PHYRE-{B, 2B} in the within-template and cross-template settings. Success percentages are averaged over all test tasks and 10 folds. Shaded regions show one standard deviation.

|  | **PHYRE-B** | | **PHYRE-2B** | | | **PHYRE-B** | | **PHYRE-2B** | |
|---|---|---|---|---|---|---|---|---|---|
|  | Cross | Within | Cross | Within |  | Cross | Within | Cross | Within |
| **RAND** | $13.0_{\pm5.0}$ | $13.7_{\pm0.5}$ | $2.6_{\pm1.5}$ | $3.6_{\pm0.6}$ | **RAND** | $6.8_{\pm5.0}$ | $7.7_{\pm0.8}$ | $2.2_{\pm1.8}$ | $3.2_{\pm0.9}$ |
| **MEM** | $18.5_{\pm5.1}$ | $2.4_{\pm0.3}$ | $3.7_{\pm2.3}$ | $3.2_{\pm0.2}$ | **MEM** | $15.2_{\pm5.9}$ | $2.7_{\pm0.5}$ | $1.9_{\pm1.6}$ | $3.4_{\pm0.3}$ |
| **MEM-O** | $22.8_{\pm5.0}$ | - | $4.9_{\pm3.1}$ | - | **MEM-O** | $20.1_{\pm5.6}$ | - | $3.8_{\pm3.2}$ | - |
| **DQN** | $36.8_{\pm9.7}$ | $77.6_{\pm1.1}$* | $23.2_{\pm9.1}$ | $67.8_{\pm1.5}$* | **DQN** | $34.5_{\pm10.2}$ | $81.4_{\pm1.9}$ | $22.4_{\pm10.0}$ | $74.9_{\pm1.7}$ |
| **DQN-O** | $56.2_{\pm10.5}$* | - | $39.6_{\pm11.1}$* | - | **DQN-O** | $58.2_{\pm10.9}$ | - | $41.6_{\pm11.7}$ | - |

(a) Area under the success-percentage curve (AUC-CESS) of five agents. Higher is better.

(b) Success percentage at $k = 10$ attempts of five agents. Higher is better.

Table 1: Comparison of the five agents on PHYRE-{B, 2B}. Mean and standard deviation on the 10 folds are reported. MEM-O and DQN-O perform best with no update in the within-template setting, making them equivalent to MEM and DQN in this case; thus, we omit their results. *Indicates an agent's AUCCESS is better than all others per the Wilcoxon one-sided test with $p=0.01$.

DQN-O, is able to reach a reasonably high AUCCESS of 56.2% on PHYRE-B, but is at just 39.6% on PHYRE-2B. This small change in the action space substantially decreases agent success. Notably, agents that perform online learning (⋆-O) substantially outperform their offline counterparts.

In Table 1b, we present the percentage of tasks that were solved within 10 attempts by each agent. This low-attempt regime is emphasized by AUCCESS and a goal of the PHYRE benchmark is to encourage research that improves results in this regime. The results are in line with prior observations and illustrate that the PHYRE-2B cross-template setting presents a significant challenge for all agents.

## 4.4 Analysis

Here, we analyze the effect of: (1) the number of actions that are ranked by agents at test time and (2) the "aggressiveness" of agent updates on the performance of online agents. In the supplement, we also ablate the deep Q-network (DQN) design. For these experiments, agents are trained and evaluated on the train and validation splits, respectively, using the first three (out of 10) folds.

**Number of actions ranked.** Figure 4 shows the AUCCESS of the RAND, MEM, and DQN agents as a function of the number of actions that are ranked by the agents at test time. We also present an OPTIMAL ranking agent that performs oracle ranking of the action set. The performance of the OPTIMAL agent suggests that ranking is a reasonable strategy: it solves all tasks in PHYRE-B and 95% of tasks in PHYRE-2B by ranking fewer than 100,000 attempts. For non-oracle agents, DQN is a much better ranker than MEM. As expected, AUCCESS increases as more actions are ranked, but eventually plateaus and sometimes decreases beyond a certain number of attempts. This is due to a lack of diversity in the rankings produced by the agents, which do not have a model of similarity between actions and may suggest multiple similar attempts when sampling of actions is fine-grained.

**Effect of online updates.** Online agents use examples obtained during both the training and testing stages. Figure 5 analyzes the effect of re-weighting both types of examples on the performance of online agents (on three folds). The results show that the AUCCESS of MEM-O is fairly independent of the weight used. The AUCCESS of the DQN-O agent does vary as a function of how many updates were performed at test time: online updates even impede DQN-O in the within-template setting.

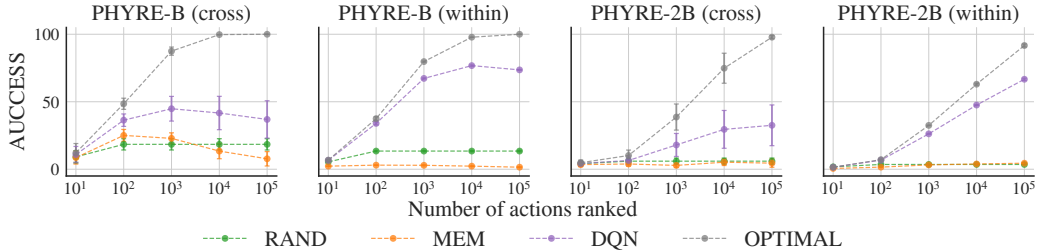

Figure 4: AUCCESS as a function of the number of actions being ranked by the agent for the RANDOM, MEM, and DQN agents and for an agent that is OPTIMAL in terms of scoring attempts.

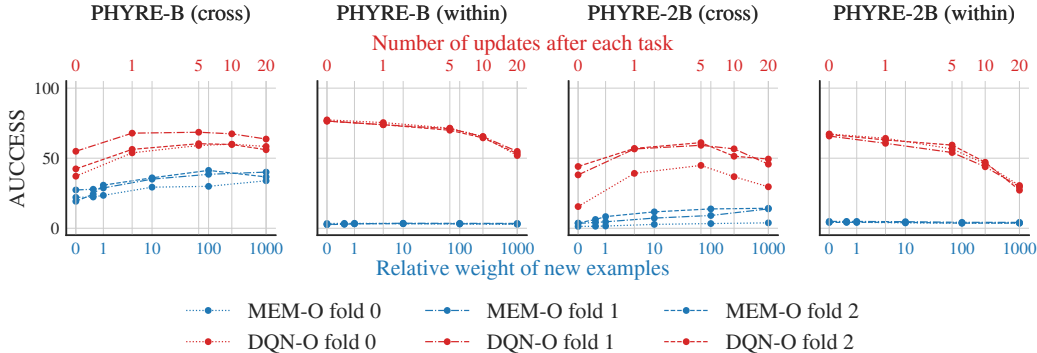

Figure 5: AUCCESS of MEM-O and DQN-O agents as the "aggressiveness" of the online update is varied during the testing phase. The left-most point in each plot is an offline version of the agent.

## 5 Discussion and Future Work

PHYRE aims to enable the development of physical reasoning algorithms with strong generalization properties mirroring those of humans [30]. Yet the baseline methods studied in this work are far from this goal, demonstrating limited generalization abilities. We foresee several areas for advancement:

- Agents should use intermediate observations from the simulator following an (unsuccessful) attempt to refine their next attempt. Our current failure to do so makes the agents sample-inefficient, as these observations contain rich information on the specific task that the agent is solving that should be used effectively for efficient problem-solving. Doing so requires *counterfactual reasoning*: agents need to reason about what would happen upon a particular change to a previous attempt.
- Agents should use a forward-prediction model that mimics the simulator by a learnable function [15]. Such a model can be integrated into a DQN by running attempts through it for a number of time steps, and using the resulting state predictions as additional inputs into the Q-network.
- Agents should explicitly diversify attempts when solving a task.
- Agents should use an active strategy at test time, *e.g.*, by starting with solving simple tasks.
- While each task is different from the others, they share the same underlying causal model (physics). Methods aimed at invariant causal prediction (ICP) [14, 38] may be well-suited for PHYRE.

Based on these observations, we expect to witness rapid progress on the PHYRE benchmark. To this point, we highlight that PHYRE is an extensible platform upon which more challenging puzzle tiers may be built. The two tiers provided in this initial benchmark are designed to be approachable, yet challenging. Future tiers may involve substantially larger and more complex action spaces.

We also foresee approaches that implement a simulator "internal" to the agent and then query it to brute-force a solution before submitting any attempts to the real simulator. Based on initial experiments, we expect that training a neural network to exactly mimic the simulator will be difficult. However, one might instead use hand-coded rules specific to PHYRE—in the extreme, one could simply call the real simulator inside the agent. We view such approaches as violating the spirit of the benchmark. We discourage this line of attack as well as in-between solutions that combine function approximation with extensive hand-coded inductive biases that are specific to PHYRE.

## Acknowledgements

We thank Mayank Rana for his help with early versions of PHYRE, and Audrey Durand, Joelle Pineau, Arthur Szlam, Alessandro Lazaric, Devi Parikh, Dhruv Batra, and Tim Rocktäschel for helpful discussions.

## Footnotes

[1]The current body vocabulary contains balls, bars, standing sticks, and jars.

[2]`https://github.com/tensorflow/models/tree/master/research/deep_contextual_bandits`

[3]To simplify follow-up research, we will release the simulation results of these 100,000 actions per task.

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
