[Supplementary Material]

# PHYRE: A New Benchmark for Physical Reasoning
## Supplemental Material

**Anton Bakhtin**     **Laurens van der Maaten**     **Justin Johnson**
**Laura Gustafson**     **Ross Girshick**
Facebook AI Research
{yolo,lvdmaaten,jcjohns,lgustafson,rbg}@fb.com

## A   Ablation Study of Deep Q-Network (DQN)

Figure 1 shows the effect on AUCCESS of six modifications to our DQN agent. The modifications encompass changes to the architecture of the action encoder (*Act1024* and *Act1024×2*), the fusion mechanism (*FuseGlobal*, *FuseFirst*, and *FuseAll*), and the balancing of training batches (*NoBalancing*); see the figure caption for full details. We make four main observations:

- Class-balancing training batches is critical to the DQN agent's performance, particularly on the PHYRE-2B tier where only 0.3% of randomly chosen actions yield a positive example.
- Early fusion of action information into the ResNet-18 observation encoder does not help. Early fusion is also inefficient for action ranking: it prohibits caching of the observation encoder's output.
- Our default fusion method uses channel-wise bias and gain modulation immediately before the ResNet-18 `conv5` stage; applying this fusion the final globally pooled features, instead, substantially deteriorates AUCCESS.
- Larger action encoders can improve performance, but the gains are not consistent across settings.

## B   PHYRE Tasks

As discussed in the main paper, the current PHYRE benchmark provides two task *tiers*: PHYRE-B tasks can be solved by placing a single ball in the initial world, whereas PHYRE-2B tasks require placement of two balls in the initial scene. Each tier provides 25 *task templates*, and each task template contains 100 *tasks* that are similar in design but that have a different initial configuration of bodies in the world. Figure 2 shows an example task from each of the 25 task templates in the PHYRE-B tier, and Figure 3 shows an example task for each of the 25 task templates in the PHYRE-2B tier.

**Stable solutions.** When designing the PHYRE tasks, we made sure that each task has a *stable solution*. We define a stable solution to be an action that: (1) solves the task and (2) still solves the task if the action is slightly perturbed. The perturbations we consider are translations by 0.5 pixels along each axis (8 shifts in total).

**Task solvability.** Because the current benchmark contains $(25 + 25) \times 100 = 5,000$ tasks, it is cumbersome to manually find stable solutions for each task. Moreover, it is not possible to do brute-force search over all possible actions because the action space is continuous. Therefore, we used the following stochastic approach to evaluate whether or not a task is solvable. Let $a$ denote an action and $\tau$ a task. We define the random variable $\texttt{stably\_solves}(a, \tau)$ to be 1 if action $a$ is a stable solution for task $\tau$ and 0 otherwise. The random variable $\texttt{valid}(a, \tau)$ is 1 iff action $a$ is a valid action for task $\tau$. We define the *solvability level* of task $\tau$ to be: $s(\tau) = P(\texttt{stably\_solves}(a, \tau) = 1 | \texttt{valid}(a, \tau) = 1)$. To determine whether task $\tau$ is solvable, we would ideally seek to reject the hypothesis $s(\tau) = 0$.

Figure 1: Mean AUCCESS on PHYRE-{B, 2B} of six DQN variants of the *Baseline* in the main text. Error bars show one standard deviation. *FuseFirst*, *FuseAll*, and *FuseGlobal* DQN agents perform fusion of observation and action features in alternative locations via channel-wise bias and gain modulation (akin to [1]): *Baseline* fuses with the input to the ResNet-18 `conv5` stage; *FuseFirst* fuses with the input to the `conv2` stage; *FuseAll* fuses with the inputs to each stage from `conv2` to `conv5`; and *FuseGlobal* fuses with the globally max-pooled output of the `conv5` stage. *Act1024* and *Act1024×2* DQN agents use *Baseline* fusion but larger action encoder networks with one or two hidden layers of 1024 units, respectively. The *NoBalancing* agents trains the *Baseline* DQN without balancing the positive and negative examples in the batches. We refer the reader to our code release on `https://phyre.ai` for full details.

Exact testing of this hypothesis is, however, infeasible, and so we resort to a proxy that uses a small constant $p_0$, randomly selected actions, and a binomial statistical test to reject at least one of the hypotheses: $s(\tau) \leq p_0$ or $s(\tau) \geq 2p_0$. We sample random actions until we can reject one of the two hypotheses. If the $s(\tau) \leq p_0$ hypothesis is rejected, we define the task to be *solvable*. Alternatively, we define the task *unsolvable* if the $s(\tau) \geq 2p_0$ hypothesis is rejected. In the unlikely event that both hypotheses are rejected we categorize the task as solvable.

It is possible to show that this algorithm requires no more than $\frac{1}{32p_0}$ action samples to reject at least one of the hypotheses with $p$-value 0.05. In practice, the value of $p_0$ was chosen to match our intuitive sense of task solvability: for PHYRE-B, we set $p_0 = 10^{-5}$; for PHYRE-2B, we set $p_0 = 10^{-6}$.

**Tier requirements.** We used the definition of task solvability to check the correctness of the implementation of a task template. We also used task solvability to guide the selection of tasks within a template, *e.g.*, the task creator may impose the constraint that a template only contains tasks with two-ball solutions and no single-ball solutions and enforce this constraint automatically.

We designed the task templates in both tiers to meet the following criteria: (1) all tasks in a tier to be solvable according to the definition of task solvability described above using samples from the action space corresponding to that tier and (2) less than 50% of the tasks in a PHYRE-2B task template can be solvable using a single ball. Hence, the task templates in PHYRE-2B are strictly harder to solve than those in PHYRE-B.

**Solution diversity.** The task templates are designed such that solving a task instance within a template should not be trivial for an agent that knows how to solve other tasks in the template. For example, a task template should not have a single "master solution" that solves (nearly) all tasks in the template. At the same time, it is nearly impossible to prevent that multiple tasks in the same template share solutions because these tasks share the same design (see Figure 4).

To measure the *solution diversity* of a task template, we count the number of tasks within the template that each action can solve. Since the action space is continuous we cannot check every action. Instead, we randomly sample $10^6$ actions to estimate solution diversity. We plot the results, for each task template, as histograms in Figure 5 and 6. Each histogram shows the number of actions ($y$-axis) that can solve a particular number of tasks ($x$-axis) within the template. We are interested to see if one or more actions are able to solve a large fraction of the tasks within a template; such actions will give

Figure 2: The 25 task templates in the PHYRE-B tier. In each task the goal is to make the (dynamic) green body touch the (static) purple body or the (dynamic) blue body; black bodies are static and gray bodies are dynamic. Each of the PHYRE-B task templates gives rise to 100 tasks, each of which can be solved by adding a single dynamic ball to the scene.

rise to bars (of any height) on the right side of the $x$-axis. The figures show that, in general, tasks in the PHYRE-2B tier require more diverse solutions to be solved than those in the PHYRE-B tier.

Figure 3: The 25 task templates in the PHYRE-2B tier. In each task the goal is to make the (dynamic) green body touch the (static) purple body or the (dynamic) blue body; black bodies are static and gray bodies are dynamic. Each of the PHYRE-2B task templates gives rise to 100 related tasks, all of which can be solved by adding two dynamic balls to the scene.

Figure 4: Each row shows five example tasks from the same task template. The size, initial position, and orientation of bodies vary within a template, so each task requires its own solution; however all tasks within a template share similar physical intuition and high-level strategy.

Figure 5: Analysis of the *solution diversity* of the task templates in the PHYRE-B tier. Histograms show the number of actions ($y$-axis) that solve a certain number of tasks in the template ($x$-axis).

Figure 6: Analysis of the *solution diversity* of the task templates in the PHYRE-2B tier. Histograms show the number of actions ($y$-axis) that solve a certain number of tasks in the template ($x$-axis).

## C Comparing Agents

To determine if one agent outperforms another agent, we use the one-sided Wilcoxon test as implemented in the `scipy.stats` Python package.[1] To enable future work to compare with our baselines, we provide AUCCESS scores for all folds and evaluation settings in Table 1.

| Setting | Fold<br>Agent | 0 | 1 | 2 | 3 | 4 | 5 | 6 | 7 | 8 | 9 |
|---|---|---|---|---|---|---|---|---|---|---|---|
| 2B (cross) | RAND | 0.0517 | 0.0212 | 0.0099 | 0.0442 | 0.0038 | 0.0356 | 0.0178 | 0.0177 | 0.0264 | 0.0275 |
| | MEM | 0.0728 | 0.0289 | 0.0135 | 0.0783 | 0.0090 | 0.0463 | 0.0186 | 0.0387 | 0.0376 | 0.0274 |
| | MEM-O | 0.0967 | 0.0371 | 0.0164 | 0.0933 | 0.0094 | 0.0815 | 0.0242 | 0.0451 | 0.0535 | 0.0345 |
| | DQN | 0.3818 | 0.1944 | 0.1072 | 0.3051 | 0.0732 | 0.2703 | 0.2388 | 0.2216 | 0.2528 | 0.2730 |
| | DQN-O | 0.5149 | 0.2682 | 0.2596 | 0.5298 | 0.2809 | 0.5313 | 0.4828 | 0.3330 | 0.3581 | 0.3987 |
| 2B (within) | RAND | 0.0271 | 0.0367 | 0.0428 | 0.0301 | 0.0394 | 0.0452 | 0.0336 | 0.0287 | 0.0380 | 0.0335 |
| | MEM | 0.0325 | 0.0336 | 0.0315 | 0.0371 | 0.0304 | 0.0314 | 0.0282 | 0.0320 | 0.0330 | 0.0347 |
| | MEM-O | 0.0325 | 0.0336 | 0.0315 | 0.0371 | 0.0304 | 0.0314 | 0.0282 | 0.0320 | 0.0330 | 0.0347 |
| | DQN | 0.6447 | 0.6829 | 0.6747 | 0.6763 | 0.6999 | 0.6700 | 0.6879 | 0.6704 | 0.6877 | 0.6824 |
| | DQN-O | 0.6447 | 0.6829 | 0.6747 | 0.6763 | 0.6999 | 0.6700 | 0.6879 | 0.6704 | 0.6877 | 0.6824 |
| B (cross) | RAND | 0.1178 | 0.1242 | 0.1818 | 0.1242 | 0.0381 | 0.2250 | 0.1173 | 0.1329 | 0.0894 | 0.1460 |
| | MEM | 0.2059 | 0.1656 | 0.2004 | 0.2263 | 0.1159 | 0.2488 | 0.1416 | 0.2467 | 0.1055 | 0.1881 |
| | MEM-O | 0.2578 | 0.2551 | 0.2443 | 0.2552 | 0.2327 | 0.2508 | 0.1469 | 0.2801 | 0.1281 | 0.2273 |
| | DQN | 0.4369 | 0.3096 | 0.4305 | 0.4391 | 0.2277 | 0.4440 | 0.3453 | 0.3920 | 0.1898 | 0.4646 |
| | DQN-O | 0.6859 | 0.4867 | 0.6671 | 0.5995 | 0.4916 | 0.6560 | 0.5100 | 0.6573 | 0.3733 | 0.4884 |
| B (within) | RAND | 0.1344 | 0.1401 | 0.1379 | 0.1380 | 0.1275 | 0.1334 | 0.1395 | 0.1430 | 0.1336 | 0.1433 |
| | MEM | 0.0198 | 0.0258 | 0.0230 | 0.0269 | 0.0223 | 0.0286 | 0.0237 | 0.0214 | 0.0223 | 0.0288 |
| | MEM-O | 0.0198 | 0.0258 | 0.0230 | 0.0269 | 0.0223 | 0.0286 | 0.0237 | 0.0214 | 0.0223 | 0.0288 |
| | DQN | 0.7682 | 0.7972 | 0.7822 | 0.7586 | 0.7703 | 0.7842 | 0.7801 | 0.7734 | 0.7804 | 0.7687 |
| | DQN-O | 0.7682 | 0.7972 | 0.7822 | 0.7586 | 0.7703 | 0.7842 | 0.7801 | 0.7734 | 0.7804 | 0.7687 |

Table 1: AUCCESS scores (on a 0.0 to 1.0 scale) of our five agents in both generalization settings, for each of our 10 folds.

## Footnotes

[1] Specifically, we call `scipy.stats.wilcoxon(A, B, zero_method='wilcox', correction=False, alternative='greater')` to test if the AUCCESS vector `A` outperforms AUCCESS vector `B`, where `A` and `B` are component-wise paired with one component for each fold.

## References

[1] E. Perez, F. Strub, H. de Vries, V. Dumoulin, and A. Courville. Film: Visual reasoning with a general conditioning layer. In *AAAI*, 2018.