[Reviews · NeurIPS 2019]

Reviewer 1



I generally like this paper. The task is compelling and the benchmark is well thought out. Experiments are well done and well reported. I think it should make a good contribution and help drive more good work, much as CLEVR has. At the same time, I have a few concerns (see Improvements below). These could be addressed in a revision and I would be interested to hear from the authors in rebuttal on these points.

Reviewer 2



I like this paper, as it presents a carefully designed benchmark for an emerging area. The initial evaluation on the benchmark also suggests future research directions. Some issues remain. First, the dataset has minimal visual complexity. It'd be great to include objects of various shapes, textures, and scenes of different lighting conditions. Note that without realistic rendering and texture, which indicate physical object properties, humans (and therefore models) may not be able to solves these tasks as well. Evaluations can be improved in two ways. - For all baselines, especially those don't work well with raw visual input (contextual bandits and policy learners), it'd make sense to use states (object and obstacle position, velocity) as input. This additional set of experiments will help to decouple visual perception from physical reasoning and highlight the merits of the dataset. - As mentioned in discussion, it's important to include some model-based RL/planning baselines. There are many papers on differentiable physics engines (e.g., interaction networks, neural physics engines). I wonder how they perform on these tasks. I failed to understand Fig 4. I'm wondering if the authors can explain what it's about and why we care about it. The authors indicated in the reproducibility checklist that links to source code and the dataset have been included, but I cannot find them.

Reviewer 3



Here are some concerns I have for this paper: - The dataset is designed to focus on "physical reasoning", which is a very broad concept. It would be better to know what aspects of physical reasoning are tested in this dataset ([40] is a good example), how are the experiment templates designed. - Following the previous point, since there is no description of the physics tested for different templates, it is difficult to draw conclusions from the results of cross-template experiments. Cross-template experiments are meaningful if they are testing similar physical reasoning processes (e.g., there could be different templates testing gravity understanding). - The selection of evaluated methods should be justified. I understand that the authors' intention is to evaluate methods that are not hand-coded with physics rules, but it seems that more appropriate baseline methods should be able to do some predictions. For example, a baseline could be a simple CNN future state prediction module combined with a success prediction module. - More like a suggestion than a criticism: it might be interesting to see human performance on this benchmark. =========================================== I have read the authors' feedback. Although I still feel that the tasks could be more clearly defined, I am fine if the paper is accepted.

[Author Response · NeurIPS 2019]

**Author Response for PHYRE: A New Benchmark for Physical Reasoning**

We thank the reviewers for their detailed and constructive comments. To recap, our submission introduces PHYRE, a
new environment for benchmarking aspects of physical reasoning in which agents are challenged to solve 2D physics
puzzles efficiently. Overall, the reviewers were positive about this contribution and liked the submission: "*I generally*
*like this paper. The task is compelling and the benchmark is well thought out.*" [R1]; "*I like this paper, as it presents*
*a carefully designed benchmark for an emerging area.*" [R2]; "*The benchmark is designed to encourage physical*
*reasoning agents that are not hand-coded.*" [R3]. The reviewers also raised concerns, which we will address next.

**[R1, R3] More analysis of reasoning skills (as in CLEVR and IntPhys).** This is a great suggestion with two
subtleties we'd like to discuss. **(1)** Analysis like that in CLEVR and IntPhys requires a taxonomy of basic reasoning
skills such that each benchmark task can be accurately described as a composition of these skills. CLEVR and IntPhys
were built after first defining their respective taxonomies. This design enables exact skill decomposition, but also
constrains the space of tasks. We intentionally took a different approach: we did not define a taxonomy upfront in
order to enable open-ended task development, which we think will lead to more diverse and challenging puzzles. Given
that, it is unclear if a post-hoc taxonomy exists for the PHYRE tasks. **(2)** A separate issue with such analysis is that it
assumes that each task can only be solved according to its prescribed skill decomposition, which is a strong assumption.
It is likely that any given task can be solved in multiple diverse ways and that some of these solutions may not involve
the reasoning skills that are assumed (e.g., by finding a creative new approach or by exploiting unidentified biases).
For example, in CLEVR it now seems likely that some models (e.g., Relation Networks) have found shortcut "cheats"
instead of using a multi-step inference process, and this outcome serves as a cautionary tale. We believe that such
in-depth analysis is a great ideal to strive for, but may not fit the open-ended nature of PHYRE tasks (if no reasonable
taxonomy exists) and may not yield the desired insights in practice (if agents find alternative solutions).

**[R1] Are two ball tasks intrinsically harder in some way that reflects interesting physical reasoning complexity?**
We think the answer is yes: solving a two-ball task requires adding two objects to the scene that must act together
in a coordinated way to achieve the goal. It is difficult to characterize what constitutes "intrinsic" difficulty, but by
any reasonable measure that we can think of (the size of the action space, the minimum number of objects involved in
the task solution, the minimum number of collisions involved in the task solution, etc.) two-ball tasks have a higher
complexity.

**[R1] Generalization beyond PHYRE?** Whether algorithms created for PHYRE will generalize to other environments
(including the real world) is a valid and important concern. Unfortunately it is not possible to know the impact of a
new dataset, environment, or benchmark ahead of time (an extreme example: it was not clear that ImageNet would
help propel the modern era of deep learning). We think that a reasonable principle to follow is: develop the simplest
benchmark that today's learning algorithms cannot solve well. As a whole, the community must "go for recall" since
a large number of potentially fruitful directions will not come to fruition and the ones that do are at times rather
unexpected in advance. We argue that if you find the task compelling and interesting, it is worth taking the risk of
promoting investigation into it.

**[R1] Reference suggestions.** Thank you for the suggestions of additional references to discuss, in particular the
workshop paper of Allan et al., which we did not yet know.

**[R2, R3] Include additional baselines: model-based learners and agents that use the world state directly.** We
agree that model-based agents are an interesting direction for exploration on PHYRE. Initially, we investigated this
direction in the form of a CNN-based forward model. We found it difficult to produce reasonable predictions more than
a second into the future and decided to abandon that exploration in favor of the methods presented in the paper. We
hope to see future work that is able to produce successful model-based learners. We also considered exposing the raw
world state to agents as the post-simulation observations, but decided given the visual simplicity of the world it should
be reasonable to expect agents to use it directly or to train scene parsing methods that can detect objects and estimate
their physical states. Ultimately, in the limited scope of an 8-page NeurIPS paper one must select a small subset of all
possible experiments. By releasing PHYRE to the public, we hope to see rapid exploration of these good suggestions.

**[R2] Minimal visual complexity; is it enough for humans?** Based on our experience playing with the tasks (and
games like Brain It On, which have a similar level of visual complexity), the provided information is sufficient.

**[R2] Fig. 4 was unclear.** We will attempt to improve the clarity. All of the agents in our experiments work by ranking
a set of $10^5$ possible solutions. This figure provides an ablation study showing how using smaller candidate solution
sets (down to just 10 candidates) influences the performance of the agents.

**[R2] Code release.** We are preparing the code release right now; it will be publicly available in the near future.

[Meta-Review · NeurIPS 2019]

The authors introduce a new game-style benchmark for physical reasoning, PHYRE, which contains a set of puzzles in a 2D physical environment using a set of parameterized task templates and variations on each template. The paper also presents baseline agents based on a non-parametric memorization strategy, DQN, and online learning variants of these agents. Reviewers are concerned that there is not enough visual complexity (shapes, textures, etc.), that the domain of physical reasoning is quite limited, and that the evaluations can be improved with more rigorous baselines. Although two reviewers see the work as marginally below threshold, all reviewers think an "accept" is reasonable.